# Incidence and predictors of mortality among road traffic accident victims admitted to hospitals at Hawassa city, Ethiopia

**Amanuel Ayele Afacho**[1]*, **Teshale Belayneh**[2], **Terefe Markos**[1], **Dereje Geleta**[1]

1 Department of Public Health, Hawassa College of Health Sciences, Hawassa, Sidama, Ethiopia, 2 School of Public Health, College of Medicine and Health Sciences, Hawassa University, Hawassa, Sidama, Ethiopia

☯ These authors contributed equally to this work.
* amanuelayele5335@gmail.com

## Abstract

### Background

Globally, road traffic accidents are the eighth-leading cause of death for all age groups. The estimated number of road traffic deaths in low income countries was more than three times as high as in high-income countries. Africa had the highest rate of fatalities attributed to road traffic accidents. Ethiopia has the highest number of road traffic fatalities among Sub-Saharan African countries. The main objective of this study was to determine the incidence and predictors of mortality among road traffic victims admitted to hospitals in Hawassa City.

### Methods

A facility-based retrospective cohort study was conducted using secondary data from hospital records. A total of 398 road traffic accident victims admitted to selected hospitals in Hawassa city from January 2019 to December 2021 participated in the study. Data were analyzed using STATA version 14.1. The Cox regression model was used to determine the predictors of mortality. A hazard ratio with a 95% confidence interval and a cut-off value of P<0.05 was used to declare the risk and statistical significance, respectively.

### Result

The incidence rate of mortality for road traffic accident victims was 7.34 per 10,000 person-hours. The predictors of mortality were the value of GCS at admission <8 (aHR = 5.86; 95% CI: 2.00–17.19), GCS at admission 9–12 (aHR = 3.27; 95% CI: 1.28–8.40), the value of SBP at admission ≤89mmHg (aHR = 4.41; 95% CI: 2.22–8.77), admission to the ICU (aHR = 3.89; 95% CI: 1.83–8.28) and complications (aHR = 5.48; 95% CI: 2.74–10.01).

**Data Availability Statement:** All relevant data are within the paper and its Supporting information files.

**Funding:** The author(s) received no specific funding for this work.

**Competing interests:** The authors have declared that no competing interests exist.

## Conclusion

The incidence of mortality among road traffic victims admitted to hospitals in Hawassa city was high. Thus, thorough follow-up and intensive management should be given to victims with critical health conditions.

## Background

A road traffic accident (RTA) is an incident on a way or street open to public traffic, resulting in one or more persons being injured or killed and involving at least one moving vehicle [1]. RTAs usually result in injury, disability, fatality, and property harm moreover as money prices to each society and therefore the people concerned [2]. There was physical, economic, social, and psychological health impact on the victim as well as families, mainly loss of productive age group as a result of road traffic accidents [3].

The World Health Organization's (WHO) 2023 report shows that 1.3 million people die and millions more are injured or disabled each year as a result of RTAs around the world. Most of the world's road traffic fatalities occur in low- and middle income countries, even though these countries have a limited number of vehicles [4]. The Africa region had the greatest rate of fatalities; 27.2 per 100, 000 population, which is the highest compared to the global rate, which was 16.7 per 100, 000 population, according to the Global Health Observatory report [5]. Sub-Saharan Africa (SSA) countries are listed in the top 20 road traffic crash-prone countries for injuries and fatalities. Ethiopia ranks second regarding RTA fatalities from east African countries [6].

Factors contributing to road traffic injury-related deaths are pre-crash, crash, and post-crash factors, which in turn are classified into individual, vehicle, and environmental level factors that can result in severe and fatal outcomes [7]. Using seat belts, helmets, and child restraints significantly decreases mortality [7,8]. Although there were global efforts to improve post-crash care in developing countries, they still lack appropriate and adequate post-crash care, contributing to the high burden of deaths and disabilities resulting from RTIs [9]. This study aimed to determine the incidence and predictors of mortality due to road traffic accidents.

## Methods and materials

### Study design and setting

A facility-based retrospective cohort study was conducted among RTA victims admitted from January 1, 2019, to December 31, 2021, in public and private hospitals in Hawassa city. The data was collected from Hawassa University Comprehensive Specialized Hospital (HUCSH) and Yanet Trauma and Surgery Specialty Center (YTSSC).

### Study period and population

The data was accessed from Feb 28 to Apr 4, 2022. All RTA victims who were admitted to hospitals and had got health care services. Study subjects were randomly selected using the victim card. Those RTA victims admitted for less than 24 hours in the facilities or referred to other health facilities with incomplete information on the variable of interest were excluded.

## Sample size and sampling technique

The sample size was calculated based on the double population formula using Epi Info version 7.2, considering the following assumptions: percent in unexposed = 23.4%, ratio (unexposed to exposed = 1, AHR = 1.6, power = 80%, level of significance = 5% and the 10% of calculated sample added for incomplete records (366*0.10 = 36.6) and added to 366+36.6 = 402.6). The final sample size was 403.

The proportional allocation of the sample was done to selected health institutions. A sampling frame or list of all RTA patients' card numbers or medical record number (MRN) who had RTAs during the study period was prepared to select study subjects by simple random sampling technique using a computer-generated random number. The victims' cards were selected randomly using simple random sampling since all populations had a sampling frame using MRN (medical record number) in the record book. The randomly selected patient cards were used to collect data.

## Data collection tools and procedure

The data collection checklist was developed based on study objectives after a thorough review of the literatures [10–12] and adapted from injury surveillance guideline document of WHO [13]. A pretested checklist was used for recording information that extracted from patient cards. Data was collected by trained Bsc clinical nurses. Data were collected on socio-demographic, pre-hospital and in-hospital factors of the victims.

Prior separation of cards which were registered twice before the beginning of data extraction has done to avoid possible duplicate data collection. The follow-up period was from the time of arrival at the hospital to the time of discharge. The first day and hour to start to follow up was the selected victim registered at the date 1st January 2019 at 2:30 am. Victims followed for 30 days whether or not the event was to occur until the discharge date [12].

The 5% of the checklist was pre-tested as well as important modifications were made. During the data collection process, the filled checklist was checked for completeness, consistency and accuracy by the supervisors and principal investigator every day. The frequency statistics values are used to check incompleteness, inconsistencies and inaccuracy depend on the measurement scale.

## Variables and measurement

The main outcome variable of the study was time to death. It is the time from the occurrence of a road traffic accident admission to the occurrence of death. RTA death: defined as any patient or victim admitted and starting treatment at selected health care facilities as a result of a road traffic accidents who lost his/her life during the course of treatment before being transferred to another health facility or discharged from the hospital. The injury severity score: is an established medical score to assess trauma severity, which correlates with mortality, morbidity, and hospitalization time after trauma [14].

Revised trauma score is a physiologic scoring system based on the initial vital sign of a victim. It is made up of GCS; SBP and RR. A low score indicates higher severity of injury [15]. Event (1): is defined as the occurrence of death from a road traffic injury. Time to death: is the time from the occurrence of the RTA admission to the occurrence of death. Censored (0): is when a victim is transferred after the study started, discharged against medical advice, study time completed or alive at the time of discharge. Glasgow coma scale: is classified as severe (GCS 3–8), moderate (GCS 9–12) and mild (GCS 13–15) [16]. Kampala trauma score II: is classified as mild injury 9–10, moderate injury 7–8 and severe injury ≤6 [17].

## Data processing and analysis

The data were cleaned and analyzed using STATA version 14.1. Descriptive statistics were used to summarize the data. The Kaplan-Meier survival curve with log-rank test was used to test for the presence of differences in the incidence of death among different categories of variables. The proportionality of hazards over time was checked using the Schoenfeld residuals plot for each covariate that crossed the zero line several times, as well as the linearity assumption, which was checked using martingale residuals [18]. The incidence of mortality was calculated. Cox's proportional hazard model was used to determine the explanatory factors. Model fitness was assessed using the Cox-Snell residual. Those variables with p<0.20 in bivariate analysis were eligible for multivariate analysis. The p-value < 0.05 was considered the threshold of statistical significance, and the 95% CI for the adjusted hazard ratio was used to measure the association between independent and outcome variables.

## Ethical consideration

Ethical clearance was obtained from Hawassa University, College of medicine and health Sciences institutional review board. Permission was obtained first from officials of selected hospitals. Data was handled confidentially in all phases of research activities.

## Results

### Characteristics of the study subject

The study was conducted on 398 road traffic accident victims. The majority (83.4%) of the study participants were male. The mean age was 32.28±15.02 years. More than eighty percent of the participants were in the productive age group (Table 1).

**Table 1. Pre-hospital conditions of RTA victims admitted to hospitals at Hawassa city; Jan, 2019 to Dec, 2021; (n = 398).**

| Variables | Category | Frequency | Percentage |
|---|---|---|---|
| Sex | Male | 332 | 83.4 |
| | Female | 66 | 16.6 |
| Age | 0–15 years | 32 | 8.0 |
| | 16–30 years | 190 | 47.7 |
| | 31–45 years | 108 | 27.1 |
| | 46–60 years | 46 | 11.6 |
| | >60 years | 22 | 5.5 |
| Victims' role (n = 398) | Pedestrian | 140 | 35.18 |
| | Passenger | 113 | 28.39 |
| | Driver | 145 | 36.43 |
| Mechanism of accident (n = 398) | Pedestrian direct hit by vehicle | 140 | 35.18 |
| | Rollover | 128 | 32.20 |
| | Collision | 130 | 32.62 |
| Referral status of the victims (n = 398) | Referred | 289 | 72.61 |
| | Self-referral | 109 | 27.39 |
| Having referral form (n = 398) | No | 109 | 27.39 |
| | Yes | 289 | 72.61 |
| Sources of referring hospitals (n = 398) | Private | 22 | 5.53 |
| | Public | 267 | 67.08 |
| | Self-referral | 109 | 27.39 |
| Presence of comorbidities (n = 398) | No | 356 | 89.45 |
| | Yes | 42 | 10.55 |

## Pre-hospital conditions of the study participants

The median time to arrive at the health facilities was 8 hours +/- 20 hours. Three fourths (72.61%) of admitted RTA victims were referred from other health facilities. All of the victims referred from health facilities were provided with a written referral form, and two-thirds (67.08%) of them were referred from public health facilities. Among the road user categories, 36.43% were drivers, followed by 35.18% pedestrians, and 28.39% passengers. Regarding the mechanism of road traffic accidents, out of the admitted subjects, 35.18% were pedestrians hit directly by vehicles, 32.62% were collisions, and 32.20% were rollovers. The majority, 89.45% of the victims, had no comorbid conditions (Table 1).

## Injury pattern and characteristics

More than one-third (35.17%) of the victims have had poly-trauma. About 9.55% of victims encountered visceral organ injury, of which nearly half (44.74%) have had an injury to the lung. About 64.82% of the road traffic accident victims were bleeding during the time of arrival to the hospital or prior, and 63.56% presented with open wounds. Nearly three-quarters (73.1%) of the accident victims had encountered bone fractures on different body parts. Among them, 56.36% were on extremities (Table 2).

## Admission and management of the victim

The mean systolic blood pressure, diastolic blood pressure, pulse rate, and respiratory rate of RTA victims at admission were 116.89 +/- 25.14 mmHg, 72.81 +/-17.44 mmHg, 9.84 +/-19.96 beats per minute, and 21.98 +/- 6.24 breaths per minute, whereas the mean value of temperature was 36.49 +/- 0.66. Injury severity was computed using the revised trauma score, and the

**Table 2. Injury characteristics of victims admitted to hospitals at Hawassa city; Jan, 2019-Dec, 2021; (n = 398).**

| Variables | Category | Frequency | Percentage |
|---|---|---|---|
| Polytrauma(n = 398) | No | 258 | 64.82 |
| | Yes | 140 | 35.17 |
| Injured body part (n = 398) | Extremities | 95 | 23.87 |
| | Head | 135 | 33.92 |
| | Multiple | 140 | 35.17 |
| | Others | 28 | 7.04 |
| Visceral organ injury (n = 398) | No | 360 | 90.45 |
| | Yes | 38 | 9.55 |
| Presence of bleeding (n = 398) | No | 140 | 35.18 |
| | Yes | 258 | 64.82 |
| Any bone fracture (n = 398) | No | 107 | 26.88 |
| | Yes | 291 | 73.12 |
| Particular fractured bone (n = 291) | Extremity | 164 | 56.36 |
| | Face | 35 | 12.03 |
| | Skull | 56 | 19.24 |
| | Multiple | 24 | 8.25 |
| | Others | 12 | 4.12 |
| Presence of open wound (n = 398) | Not present | 149 | 37.44 |
| | Present | 249 | 63.56 |

Note; Multiple means more than two body parts or bone fractures

**Table 3. Admission characteristics of RTA victims admitted hospitals at Hawassa city; Jan, 2019 to Dec, 2021; (n = 398).**

| Variables | Category | Frequency | Percentage |
|---|---|---|---|
| GCS at admission (n = 398) | <8 GCS value | 34 | 8.55 |
| | 9–12 GCS value | 66 | 16.58 |
| | 13–15 GCS value | 298 | 74.87 |
| SBP at admission | >89 mmHg | 351 | 88.19 |
| | ≤89 mmHg | 47 | 11.81 |
| History of LOC (n = 398) | No | 204 | 51.26 |
| | Yes | 194 | 48.74 |
| Reason for admission (n = 398) | Surgery & resuscitation | 329 | 82.69 |
| | Close observation | 69 | 17.31 |
| Admission to ICU (n = 398) | Yes | 52 | 13.06 |
| | No | 346 | 86.94 |
| Victims' outcome (n = 398) | Improved | 320 | 80.40 |
| | Worsened | 60 | 15.07 |
| | Others | 18 | 4.03 |
| Injury pattern in years (n = 398) | 2019 | 91 | 22.86 |
| | 2020 | 131 | 32.92 |
| | 2021 | 176 | 44.22 |

Note; GCS, Glasgow coma scale; ICU, intensive care unit; LOC, loss of consciousness; SBP, systolic blood pressure

median revised trauma score was 7.55 +/- 0.86 (IQR 6.6 7.5). About 74.87% of the victims registered mild head injuries concerning the Glasgow coma scale category. The proportion of deaths concerning the GCS category was 61.76% among severe head injury victims.

Nearly half (48.74%) of road traffic accident victims had a history of loss of consciousness, and 82.69% of the victims admitted for their condition required surgery or resuscitation. Due to the severity of their injuries, 13.06% of the victims were admitted to the intensive care unit. (Table 3) more than a quarter (30.90%) of road traffic accident victims had got fixation as a management procedure. There were 12.31% of road traffic accident victims who developed hospital acquired diseases, of which 7.28% developed pneumonia and 4.07% developed complications. Three-fourths (75.38%) of the victims get both surgical and conservative management (Table 4).

## Incidence of mortality

A total of 398 patients were followed for a range of 24 to 720 hours and a maximum, with median follow-up time of 120 +/- 144 hours. During the follow-up period, 84.9% of patients were cured and discharged, 12.6% have died, and 3.5% were referred or discharged against medical advice. The total time at risk for 398 was 68118 person-hours, with an overall incidence of mortality among admitted RTA victims of 7.34 per 10,000 person-hours of observations (95%CI: 5.56, 9.68 per 10,000 person-hours of observations).

The incidence of mortality in the first (2019), the second (2020) and the third (2021) year was 7.16/10, 000, 4.24/10, 000, and 9.22/10, 000 person-hours of Observation, respectively. The incidence of mortality for the male population was higher (8.02/10,000; 95%CI: 5.9, 10.8/10,000 person-hours observation) than for females, and it was higher (10.0/10,000; 95%CI: 5.5, 18.1/10,000 person-hours observation) for those in the age group of >50 years.

Out of the accident victims admitted to both hospitals; 12.6% had died and 87.4% had survived, or their outcome was not known or censored. Among the victims who died, 12% were

**Table 4. Management and diseases among victims admitted hospitals at Hawassa city, Jan; 2019-Dec, 2021; (n = 398).**

| Variables | Category | Frequency | Percentage |
|---|---|---|---|
| Common procedures are done (n = 398) | Wound repair | 80 | 20.10 |
| | Plaster of paris | 12 | 3.02 |
| | Craniotomy | 29 | 7.29 |
| | Chest tube | 11 | 2.76 |
| | Ventilation | 45 | 11.31 |
| | Fixation | 123 | 30.90 |
| | Medication | 54 | 13.57 |
| | Others | 44 | 11.06 |
| Hospital acquired disease (n = 398) | No | 349 | 87.69 |
| | Yes | 49 | 12.31 |
| Particular disease acquired (n = 49) | Pneumonia | 29 | 7.28 |
| | Others | 20 | 5.03 |
| Complication developed (n = 398) | No | 342 | 85.93 |
| | Yes | 56 | 14.07 |
| Management given (n = 398) | Conservative | 98 | 24.62 |
| | Surgical & conservative | 300 | 75.38 |

female and 88% were male. The incidence of mortality varies among head injury categories: 1.8 per 10,000; 17.1 per 10,000; and 31.6 per 10,000 person-hours of observation for victims regarded as having mild, moderate, and severe head injuries, respectively (Fig 1).

The mortality among self-referred victims was higher than that of those who were referred from other health facilities (11.6/10,000; 95% CI: 7.3, 18.4/10,000 person-hours observation). The incidence of mortality was highest (12.9/10,000; 95%CI: 5.4, 30.9/10,000 person-hours observation) among early arrivers (<1hr) to the health facility as well as in the final year of the study (9.2/10,000; 95%CI: 6.4, 13.2/10,000 person-hours observation). The incidence of

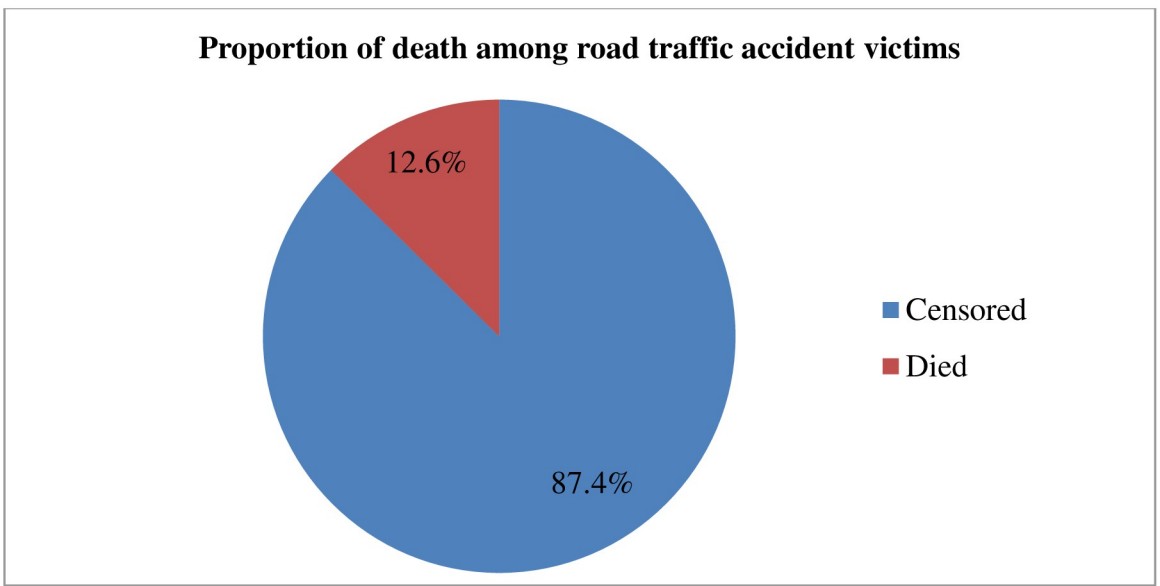

**Fig 1. Proportion of RTA victims' status admitted to hospitals in Hawassa city; Jan 2019-Dec, 2021.**

mortality was highest (11.8/10,000; 95% CI: 7.5–18.5/10,000 person-hours observation) in YTSSC as compared to HUCSH. Those who have not had referral forms had higher incidence of death (11.1/10,000; 95%CI: 6.9, 17.6/10,000 person-hours of observation) who as compared to those who have had referral forms.

Regarding mortality among each variable, victims with poly-trauma have had a high incidence of mortality (13.4/10,000 person-hours of observation, 95% CI: 9.5, 18.7/10,000). The incidence of mortality among victims with visceral organ injury was high (13.7/10,000 person-hours of observation, 95% CI: 7.4 25.4/10,000). The victims without bone fracture showed high incidence mortality (14.9/10,000 person-hours of observation, 95% CI: 9.7, 22.9/10,000), which might be because nearly half the fractured bone was on the extremities of the victim.

The incidence of mortality for those who had a history of loss of consciousness was 13.2/10,000 person-hours observation (95% CI: 9.6, 18.0/10,000); for those who have had hospital-acquired diseases was 14.6/10,000 person-hours observation (95% CI: 8.8, 24.2/10,000); and for whom merely conservative management was given was 17.9/10,000 person-hours observation (95% CI: 12.0, 26.8/10,000), which was so high.

The median survival was not computed because more than 61% of participants survived beyond the study time; rather, we calculated the cumulative and mean survival times. The mean survival time was 171.15 hours (7.11 days), with a standard deviation of 166.40 hours (6.94 days).

The cumulative proportion of patients surviving at the end of the first 24 hours after admission or injury was 95.79% (95% CI: 93.22%, 97.40%). and similarly, it was 87.54% (95% CI: 83.11%, 90.88%), 82.65% (95% CI: 77.08%, 86.98%), 78.72% (95% CI: 0.14%, 85.10%), and 61.08% (95% CI: 33.29%, 80.18%) at the end of 120 hours (5th), 240 hours (10th), 480 hours (20th), and 720 hours (30th) of admission after injury to the hospitals, respectively (Fig 2).

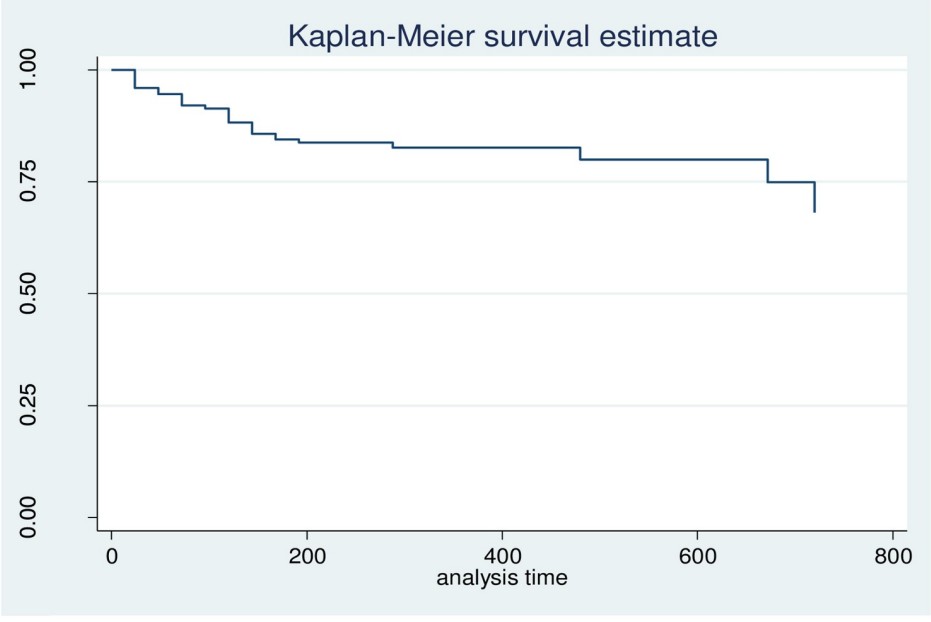

**Fig 2. Overall Kaplan Meier estimation their survival among RTA victims admitted to hospitals at Hawassa city from Jan 1, 2019 to Dec 2021.**

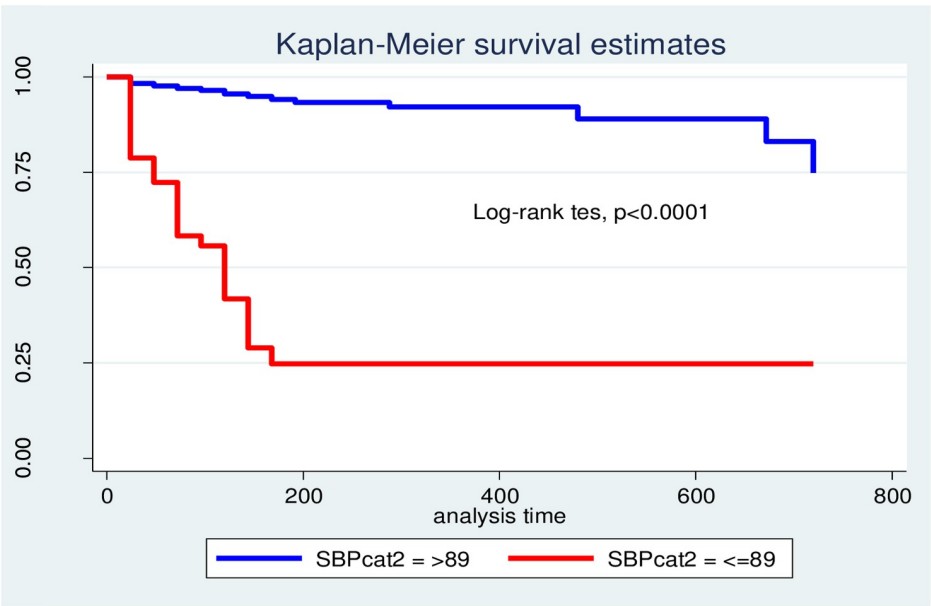

**Fig 3. Kaplan Meier survival estimate for categorized SBP among RTA victims admitted to hospitals at Hawassa city, Jan, 2019–Dec, 2021.**

The incidence of mortality among road traffic accident victims with SBP ≤89mmHg was significantly different and higher than those of SBP >89mmHg; similarly it was also true in case of admitted to ICU (Fig 3).

At 24 hours of follow-up, the probability of survival for those RTA victims with SBP≤89mmHg and >89mmHg was 78.72% and 98.20%, respectively. There was a statistically significant difference in the death of accident victims with systolic blood pressure measurement at admission >89mmHg as compared to those with ≤89mmHg. There is also a statistically significant difference in mortality between the two categories of poly-trauma.

Kaplan-Meier survival curves together with the log-rank test were fitted to test for the occurrence of death among categorical explanatory variables. There was a high incidence of mortality among victims with decreased blood pressure measurement (log-rank test, $\chi 2$ 150.12) and admission to the ICU (log-rank test, $\chi 2$ 96.44) (Table 5).

## Predictors of mortality

The variables included in the bivariate analysis were sex, age, SBP, GCS, RR, temperature values at admission, RTS, the victim's role, the mechanism of an accident, the duration to arrive at the hospital, poly-trauma, injury to a visceral organ, the history of LOC, the reason for admission, admission to ICU, the history of comorbidities, hospital-acquired diseases, complications developed, and management given. The variables that had a p-value below 0.25 in the bivariate Cox proportional hazard model as well as being important in the clinical setting were included in the multivariate analysis. Finally, the predictors of mortality in the final model were the value of GCS at admission <8 (aHR = 5.86; 95% CI: 2.00–17.19), GCS at admission 9–12 (aHR = 3.27; 95% CI: 1.28–8.40), the value of SBP at admission <89mmHg (aHR = 4.41; 95% CI: 2.22–8.77), and admission to ICU (aHR = 3.89; 95% CI: 1.83–8.28), and complication developed (aHR = 5.48; 95%CI: 2.74–11.01) (Table 6).

**Table 5. Comparison of mortality among categories of predictor variables by using log-rank test of victims admitted to hospitals at Hawassa city; Jan, 2019-Dec, 2021 (n = 398).**

| Variables | Category | IMR/10,000 pho | Log-rank (χ2) | p-value |
|---|---|---|---|---|
| Admission to ICU (n = 398) | Yes | 29 | 96.44 | <0.0001 |
| | No | 3.5 | | |
| Hospital acquired disease (n = 398) | No | 6.1 | 12.80 | 0.0003 |
| | Yes | 14.6 | | |
| Complication developed (n = 398) | No | 3.8 | 72.74 | <0.0001 |
| | Yes | 31.2 | | |
| Management given (n = 398) | Conservative | 17.9 | 21.6 | <0.0001 |
| | Surgical & conservative | 4.7 | | |
| Duration to arrive hospital (n = 398) | <1 hours | 12.8 | | |
| | 1–4 hours | 7.2 | | |
| | 4–24 hours | 7.7 | | |
| | >24 hours | 4.4 | 2.46 | 0.4817 |
| Polytrauma (n = 398) | No | 5.3 | 7.67 | 0.0056 |
| | Yes | 11.06 | | |
| History of LOC (n = 398) | No | 2.6 | 23.28 | <0.0001 |
| | Yes | 13.2 | | |
| Availability of referral form (n = 398) | No | 11.6 | 3.65 | 0.0559 |
| | Yes | 6.1 | | |
| SBP at admission (n = 398) | >89mmHG | 3.2 | 150.12 | <0.0001 |
| | ≤89mmHg | 57.8 | | |

NB: SBP, systolic blood pressure; ICU, intensive care unit; IMR, incidence of mortality; pho, person-hours observation.

The hazard of death increased by six fold [aHR = 5.86; 95% CI: 2.00–17.19] for those RTA victims whose GCS value <8 as compared to those having a GCS value of 13–15, and similarly, the hazard of death was 3 times higher for RTA victims with a GCS value of 9–12 at admission. Those victims whose SBP value at admission ≤89mmHg 89 mmHg had a 4 times higher hazard of death [aHR = 4.41; 95% CI: 2.22–8.77] as compared to those with a SBP value >89mmHg. The risk of death for those admitted to the ICU was 4 times [aHR = 3.89; 95% CI: 1.83–8.28] higher than that for those not admitted to the ICU and finally, the hazard of dying for those RTA victims who developed complications was increased by 6 fold [aHR = 5.48; 95% CI: 2.74–11.01] as compared to those who had not developed a complication (Table 6).

The cox-proportional hazard assumption was checked by using the overall Schoenfeld global test for the full model, and it was met (p = 0.4613). All covariates are met for the proportional hazard assumption. Residuals were checked using the goodness-of-fit test for Cox-Snell residuals. The final model fits the data as shown in the figure below, in which the hazard function follows 450 (Fig 4).

## Discussion

The current study found that 12.6% of road traffic accident victims died in the follow-up period, which is comparable to a study done in Ethiopia at 12.9% [19]. It is lower than the study conducted in D.R. Congo (19.6%) [20], and Tanzania (17.5%) [21]; on the other hand, this finding is higher than studies done in Kenya (7.7%) [22], Rwanda (9.4%) [23], and Ethiopia (9.5%) [24].

**Table 6. Bivariate and multivariate cox regression analysis for independent predictors of time to death among RTA victims admitted to the hospital at Hawassa city; Jan, 2019–Dec, 2021, (n = 398).**

| Variables | Category | CHR (95% CI) | AHR (95% CI) | P-value |
|---|---|---|---|---|
| Availability of referral form(n = 398) | No | 1 | 1 | |
| | Yes | 0.58 (0.32, 1.03) | 0.59 (0.29, 1.18) | 0.136 |
| Mechanism of injury (n = 398) | Pedestrian direct hit by vehicle | 1 | 1 | |
| | Rollover | 0.75 (0.39, 1.42) | 1.74 (0.82, 3.69) | 0.147 |
| | Collision | 0.55 (0.67, 4.46) | 1.12 (0.48, 2.61) | 0.784 |
| SBP at admission (n = 398) | >89 mmHg | 1 | 1 | |
| | ≤89 mmHg | 14.73 (8.29, 26.18) | 4.41 (2.22, 8.77) | <0.001 |
| History of LOC (n = 398) | No | 1 | 1 | |
| | Yes | 4.64 (2.32, 9.28) | 0.68 (0.27, 1.68) | 0.406 |
| Admission to ICU (n = 398) | No | 1 | 1 | |
| | Yes | 10.0 (5.68, 17.74) | 3.89 (1.83, 8.28) | <0.001 |
| Hospital acquired diseases (n = 398) | No | 0.34 (0.18, 0.64) | 1.64 (0.78, 3.42) | 0.189 |
| | Yes | 1 | 1 | |
| Complication developed (n = 398) | No | 1 | 1 | |
| | Yes | 7.60 (4.36, 13.27) | 5.48 (2.74, 11.01) | <0.001 |
| Management given (n = 398) | Conservative | 1 | 1 | |
| | Surgical & conservative | 0.29 (0.17, 0.51) | 0.71 (0.37, 1.34) | 0.294 |
| Polytrauma(n = 398) | No | 1 | 1 | |
| | Yes | 2.14 (1.22, 3.72) | 1.11 (0.58, 2.11) | 0.761 |
| Duration to arrive hospital (n = 398) | | 0.99 (0.97, 1.03) | 0.99 (0.98, 1.01) | 0.715 |
| GCS value at admission (n = 398) | 13–15 | 1 | | |
| | 9–12 | 9.35 (4.26, 20.54) | 3.27 (1.28, 8.40) | 0.014 |
| | <8 | 19.91 (9.09, 43.62) | 5.86 (2.00, 17.20) | 0.001 |

NB: SBP, systolic blood pressure; GCS, Glasgow coma scale; ICU, intensive care unit; LOC, loss of consciousness; CHR, crude hazard ratio; AHR, adjusted hazard ratio;

This study indicated the incidence of mortality among road traffic accidents was 7.34 per 10,000 person-hours of observations. This result is higher when compared to studies conducted in Iran 4.51 per 10,000 [25], and Ethiopia 2.9 per 10,000 and 1.0 per 10,000 [12, 26]; the possible explanation for this disparity could be that our study was merely conducted on cases admitted to the hospital at least for 24 hours that signal severity of injury to victims. Another might be the difference between hospitals in their emergency management and critical care set-ups.

Our study revealed that the male population was mostly affected by road traffic accidents; the ratio of male to female was 5:1. This finding is in line with studies conducted in Rwanda, the D.R. Congo, and Kenya [20, 22, 23] and the ratio is consistent with the studies done in Iran (4:1) and Nepal (6:1) [27, 28]. The proportion of deaths in this study is higher in males than the females, which is again, in line with studies conducted in Romania and Thailand [29, 30].

In our study, the incidence of mortality among the male population (8.0 per 10,000 person-hours of observation) was higher than that among female population (4.5 per 10,000 person-hours of observation) in our study. This finding coincides with the study conducted in Iran (male 7.2/10,000 vs. female 1.6/10,000), because the male population is more involved in outside work than the female population, including travel by using automobiles for distance travel [25].

The current study revealed that the incidence of mortality among road traffic accident victims in the age group above 50 years was high, which is not consistent with studies undertaken

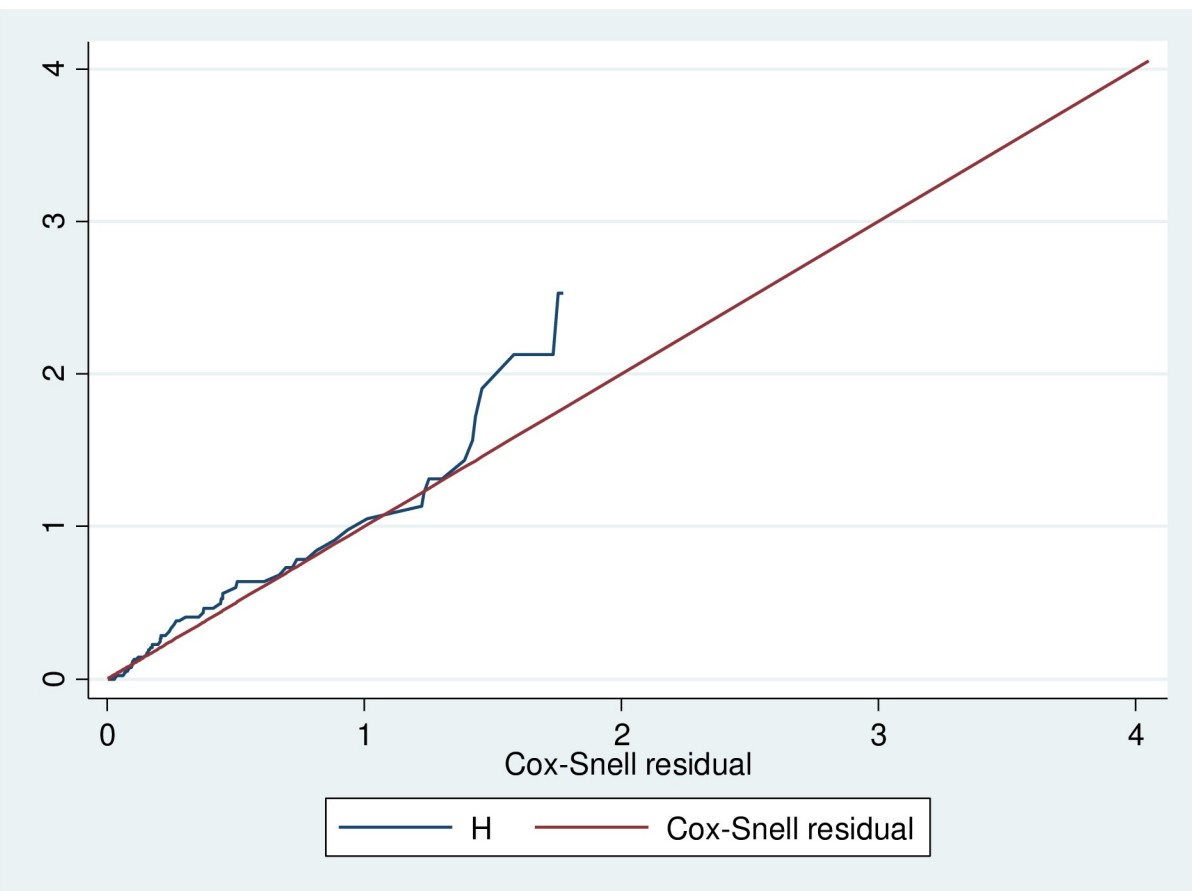

**Fig 4. Nelson-Aalen cumulative hazard graph against Cox-Snell residual on road traffic accident victims admitted to hospitals at Hawassa city from Jan 1, 2019-Dec 31, 2021.**

in Yemen, Ghana, the D.R. Congo, and Ethiopia where the rate of death was higher among the age group below 50 Years [12, 20, 21, 25]. The possible reason behind this is the fact that the older age group might have decreased immunity and comorbidities, making them more susceptible to nosocomial infection and unable to respond to treatment.

According to our study, the incidence of mortality was high (12.86 per 10,000 person-hours of observation) among early arrivals (<1 hour) to the hospital. This result is contrary to studies done in India, Tanzania, and Ethiopia [31–33] which showed that late arrival increases mortality among road traffic victims. It is also contrary to the golden time concept for the management of emergency victims [34]. The possible explanation for the discrepancy might be that the road traffic accident victims who arrived early were those with severe injuries.

Polytrauma victims have had high (13.4 per 10,000 person-hours of observation) mortality as compared to victims with mono-trauma in our study which is associated with mortality (p<0.0001). The finding is in agreement with studies undertaken in Tanzania and Iran, which confirmed that the presence of polytrauma was an independent factor in increasing the risk of death among road traffic injury victims [32, 35].

Our study confirmed that those who have been admitted to the ICU have a higher risk of death than those who have not been admitted to the ICU. The finding is not consistent with studies done in Australia and Kenya, which reported that admission to an ICU is protective

[22, 36]. The possible explanation for the discrepancy could be late admission to the ICU due to the scarcity of beds and patient load. Another possible reason might be most of the study victims who arrive at these hospitals spent their time in another institution without getting adequate treatment.

Low systolic blood pressure at admission was a significant predictor for mortality among road traffic accident victims in our study. This finding is in agreement with previous studies that confirmed victims with low systolic blood pressure were at increased risk of death, which might be due to acute blood loss [12, 21, 26, 36]. Poor blood perfusion to the vital organs due to low systolic blood pressure leads to a decompensation state with multiple organ failure and could increase mortality among road traffic accident victims [37].

According to our study, the low Glasgow coma scale increases the hazard of death among road traffic accident victims as compared to those with a higher GCS [10–12] at admission. This finding was supported by different studies conducted in Australia, Kenya, and Ethiopia, i.e.; those victims who had a low GCS value were at a higher risk of death than those who had a high GCS value at admission [22, 31, 36]. And also in our studies, victims with severe head injuries (GCS≤8) had the highest incidence of mortality, which is in line with other studies conducted in Ethiopia [12, 38].

Further in line with the GCS value, those with severe head injuries were at higher risk to die following road traffic as compared to those with mild head injuries. This finding is in agreement with studies conducted in Tanzania, the Democratic Republic of the Congo (DRC), and Kenya that verified those RTA victims who have had severe head injuries were at increased risk of death compared to those who had suffered a mild head injury [20–22].

Our study found that developing complications during admission was a predictor of mortality among road traffic victims, which meant the hazard of death increased for those who developed complications as compared to their counterparts. This finding is in line with studies conducted in America and Germany. The complications might be in different parts of the body, which in turn affects vital system functions that lead to death in road traffic accident victims [39–42].

## Limitations of the study

As this study is retrospective, the findings of our study were affected due to missing variables or variables not recordable in the facilities. be, in turn, of Our study does not include deaths that occurred at home since those victims who were discharged against medical advice and referred to other health facilities for further investigation and management were at increased risk of death. It was so difficult to include those factors that affect the mortality of road traffic accident victims due to the nature of the card review, like pre-hospital care, distance from the hospital, and others that may be confounders.

## Conclusion

The incidence of mortality among road traffic victims admitted to hospitals in Hawassa city was high, with more than half of all deaths occurring within 72 hours of admission. The predictors of mortality among road traffic accident victims were having low systolic blood pressure, a low Glasgow coma scale, being admitted to the intensive care unit, and developing complications. Health facilities should provide the necessary resources and equipment for emergency management; recruit skilled professionals in emergency management; and provide the job training to fill the skill gap. Healthcare workers should give special attention to victims with low blood pressure and a lower Glasgow coma scale who were admitted to the ICU and developed complications; they should strictly follow emergency management and critical care

protocols and work as per protocol. Researchers should further study the area of intensive care units for road traffic accidents due to the retrospective nature of our study and the effect of prior hospital management on the outcome of the study.

## Supporting information

**S1 File.**
(ZIP)

## Acknowledgments

The authors would like to acknowledge Hawassa University and Hawassa College of Health Sciences for providing support to conduct this study. We also would like to express our deepest appreciation and gratitude to Hawassa University Comprehensive Specialized Hospital and Yanet Trauma Surgical Specialty Center for their unreserved support in providing information for the successful accomplishment of this study. Finally, we would like to express our appreciation to all who are in favor of our properly conducting research.

## Author Contributions

**Conceptualization:** Amanuel Ayele Afacho.

**Data curation:** Amanuel Ayele Afacho.

**Funding acquisition:** Teshale Belayneh.

**Methodology:** Amanuel Ayele Afacho, Teshale Belayneh, Terefe Markos, Dereje Geleta.

**Supervision:** Terefe Markos, Dereje Geleta.

**Writing – original draft:** Amanuel Ayele Afacho.

**Writing – review & editing:** Amanuel Ayele Afacho, Teshale Belayneh, Terefe Markos, Dereje Geleta.

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
