## [Decision Letter · Decision Letter 0]

29 Aug 2023

PONE-D-23-20417Incidence and predictors of mortality among road traffic accident victims admitted to hospitals at Hawassa city, EthiopiaPLOS ONE

Dear Dr. Afacho,

Thank you for submitting your manuscript to PLOS ONE. After careful consideration, we feel that it has merit but does not fully meet PLOS ONE’s publication criteria as it currently stands. Therefore, we invite you to submit a revised version of the manuscript that addresses the points raised during the review process.

Specifically:Obviously a great effort has been done to obtain all datas. However, the manuscript is not yet ready for publication. Both reviewers are experts in the field and by the way native english speakers. They made recommendations to improve your manuscript. Please follow these suggestions and, above all, consult a native english speaker to achieve standard english.

We look forward to receiving your revised manuscript.

Kind regards,

Hans-Peter Simmen, M.D., Professor of Surgery

Academic Editor

PLOS ONE

http://etd.hu.edu.et/bitstream/handle/123456789/2817/Amanuel%20final%20thesis.pdf?isAllowed=y&sequence=1

In your revision ensure you cite all your sources (including your own works), and quote or rephrase any duplicated text outside the methods section. Further consideration is dependent on these concerns being addressed.

Reviewers' comments:

Reviewer's Responses to Questions

**Comments to the Author**

1. Is the manuscript technically sound, and do the data support the conclusions?

Reviewer #1: Partly

Reviewer #2: Yes

2. Has the statistical analysis been performed appropriately and rigorously? 

Reviewer #1: Yes

Reviewer #2: I Don't Know

3. Have the authors made all data underlying the findings in their manuscript fully available?

Reviewer #1: Yes

Reviewer #2: Yes

4. Is the manuscript presented in an intelligible fashion and written in standard English?

Reviewer #1: No

Reviewer #2: Yes

5. Review Comments to the Author

Reviewer #1: This investigation analyses the incidence and predictors of motility among road traffic accidents in Ethiopia. I have several recommendations and remarks.

"Road traffic accidents had a tremendous impact on the physical, economic, social, and psychological health of the victim as well as families, mainly loss of productive age group [3]"

Please rephrase the sentence.

"The World Health Organization's (WHO) 2018 report shows that 1.35 million people die and 20 to 50 million more are injured each year as a result of RTAs around the world. More than 90% of the world’s road traffic fatalities occur in low- and middle-income countries, even though these countries have a limited number of vehicles [4]"

This reference is 5 years old. Please provide a more recent analyzation of the world health organization.

"Factors contributing to road traffic injury-related deaths are pre-crash, crash, and post crash factors, which in turn are classified into individual, vehicle, and environmental level factors. These crash factors result in severe and fatal outcomes."

Please provide a references to this this statement.

"All RTA victims who were admitted to hospitals and who got health care services there were included in the study. Study subjects were randomly selected using the victim card. Those RTA victims admitted for less than 24 hours in the facilities or referred to other health facilities with incomplete information on the variable of interest were excluded."

Please clearly state inclusion criteria and exclusion criteria for the study population. Whether mono trauma is such as distal radius fracture part of this investigation?

"To study the victims’ cards were selected randomly using simple random sampling since all populations have a sampling frame using MRN in the record book. The randomly selected record books/registries/database/patient cards were used to collect data. The proportional allocation of the sample was done"

Pleased clearly state why these selection process was carried out in such manner. This random collection of patients makes the data hard to compare with now established exclusion and inclusion criteria. This needs to be fully addressed in the discussion and limitation section.

"MRN" please define this abbreviation.

"WHO[13]"

This reference is 22 years old. Please explained in great detail, the advantages and disadvantages of this study design and this protocol chosen.

"important modifications were made"

With kind of modifications were made and why?

"398 road traffic accident victims" there is a discrepancy in the previous reported 403 included patients to this statement. Where are the 5 missing patients?

"The majority, 89.45% of the victims, had no comorbid conditions. (Table 1)"

What kind of comorbid conditions where these? Are they contributing to the outcome? Such as blood-thinners are pre-existing severe illnesses?

"More than one-third (36.93%) of the victims have had injuries to multiple parts of the body. whereas 35.17% were injuries to multiple body regions".

Please clearly state the difference here again, so the reader can follow.

"About 64.82% of the road traffic accident victims had to bleed"

Please rephrase the sentence.

"The most commonly performed management procedure was fixation (30.90%)"

To get a better understanding of the mortality rate, it is important to differentiate between the first surgical treatment of the patient from the external referring hospital and the receiving hospital. For example, when were the interventions carried out? Since the majority of the patients were transferred from different hospitals, the need to be a distinction of when the surgical interventions were carried out.

Results:

The results are really extensive and should be reduced to the main findings of the study. Again there should also be a distinction between patients who received initial treatment in outside hospitals and patients who have been directly admitted to the hospital where the study was conducted.

In the described mortality rate, what was the leading cause of death.? It is described in great detail, that the GCS values and a high impact on mortality. However accompanying injuries and leading cause of death is not clear to me.

"The current study revealed that the incidence of mortality among road traffic accident victims in the age group above 50 years was high, which is not consistent with studies undertaken in Yemen, Ghana, the D.R. Congo, and Ethiopia where the rate of death was higher among the age group below 50 years"

Please add citation.

"The possible explanation for the discrepancy might be that the road traffic accident victims who arrived early were those with severe injuries."

Please explain additional interpretation of the data. Late arrival to the hospital usually resolves and higher mortality rate. Of course there is a Biers Cynthia really injured patient died before they can reach the hospital due to delayed treatment. Therefore it is important to know and to distinguish between initial outside treatment of the patient, treatment from the paramedic due to the transport etc.

"Another possible reason might be that most of the study victims who arrive at these hospitals spent their time in another institution without getting adequate treatment."

Again this is exactly the point that need to be discussed and the need to be the focus of the fairly further discussion.

In general the discussion section is really extensive and need to be shortened by at least half. Some findings are obvious and well known predictors for mortality. These can be summarised in one paragraph. The unusual findings need to be discussed and addressed with findings from the literature

Conculsion:

Conclusions too long and to generally speaking. The real benefit of this studies shld be to identify the main trauma for specific accidents. Then a scientific recommendation could be given for preventive measures as well as treatment measures of these patients in order to minimalize the high mortality rate.

Reviewer #2: Thank you for the opportunity to review this interesting study. This study was a retrospective cohort study to evaluate mortality rate in road traffic accidents in Hawasa city, a city in south central Ethiopia. Patients who suffered a road traffic accident and were admitted at two trauma hospitals for at least 24 hours were randomly selected and included in the study. The effort by the authors to gather and analyze a large data set is clearly evident. Further, the data is presented clearly in multiple tables. I appreciate the effort of the authors in clearly stating the main findings of the study.

However, some of the simple demographic data does not make sense. The time to admission and length of hospital stay, for example. Here median is sometimes presented with SEM, where IQR should be stated. Further some of SEM presented don’t make sense. Please reevaulate and correct where necessary.

Follow up should be called length of hospital stay up to the maximum study period of 30 days. The study had no follow-up period after patient discharge from the hospital.

Could the authors please shortly explain the rationale for using patient hours of observation as the metric for conveying mortality rate? I am not too familiar with this presentation, as it is difficult to evaluate clinically and I would appreciate a bit more insight here. A much more interesting and commonly used metric would be listing mortality rates at certain time intervals since hospital admission and/or time from the injury itself. For example, presenting mortality rates at specific time intervals, as was described in the results section, and is based on the Kaplan Meier curve in Figure 2 is very nice and is a clinically more pertinent way of presenting the mortality data in this study. I would recommend adding this data in table form to make it more easily understood.

It would be interesting to specify mortality rates based on which of the major visceral organs (ie lung, liver, spleen, intestines) was injured, as opposed to grouping injuries of all visceral organs together. Also please explain the discrepancy in the abstract between n=403 and n=398 in the study results.

Please refrain from analysing study results in the results section. Analysis and interpretation of the results should be restricted to the discussion section of the paper. Also, which variables were assessed in bivariate analysis should be included in the method section, and not in the results section. In the results section, the data from Table 6 should be presented. Further, in Table 1, please remove “having removal form” as this is redundant since referral status was also listed. In Table 5, years are listed, I assume you mean “hours.” Please correct or elaborate. In Table 6, please provide an extra column with the overall mortality rate as the absolute values are presented nicely here.

Finally, a short statement about the increase in patients in 2021 versus 2019 would be interesting. This may have been due to COVID 19, as we saw similar, marked decreases in hospital admissions due to injury in 2019 and also to a lesser extent in 2020. If not due to COVID, please elaborate.

This study does have several limitations that were addressed by the authors and are common in studies like this. A further important limitation is the relatively short study period of 30 days. Please state how many patients were still hospitalized at 30 days, as very long hospital stay is also a known risk factor for death and may have skewed study results. This is supported by the Kaplan Meier curve presented in Figure 2 that shows a sharp decrease in survival around the end of the study period.

Also, while the paper is understandable in its present form, I would recommend having a native English speaker edit the manuscript for better comprehension.

6. PLOS authors have the option to publish the peer review history of their article (what does this mean?). If published, this will include your full peer review and any attached files.

Reviewer #1: No

Reviewer #2: No

---

## [Author Response · Author response to Decision Letter 0]

5 Dec 2023

Comments to the author section (Reviewer #1)

1. Is the manuscript technically sound, and do the data support the conclusions?

Yes 

2. Has the statistical analysis been performed appropriately and rigorously?

Yes 

3. Have the authors made all data underlying the findings in their manuscript fully available?

Yes 

4. Is the manuscript presented in an intelligible fashion and written in Standard English?

Yes 

5. "The World Health Organization's (WHO) 2023 report shows that 1.3 million people die and 20 to 50 million more are injured each year as a result of RTAs around the world. More than 90% of the world’s road traffic fatalities occur in low- and middle-income countries, even though these countries have a limited number of vehicles [4]"

Changed to recent global data

6. Please clearly state inclusion criteria and exclusion criteria for the study population. Whether mono trauma is such as distal radius fracture part of this investigation?

Our study outcome variable was occurrence of mortality among road traffic accident victims admitted to hospitals. Therefore; we were not differentiating between affected body parts since we followed all accident victims with in study period for event (death).

We were also studied predictors of mortality (event) with in study period among road traffic accident victims. The affected body part might had significant impact on mortality of victims, that means those having had head injury could be more prone to death as compared to having had extremity injury though we included all victims of road traffic accident. 

7. "WHO [13]" This reference is 22 years old. Please explained in great detail

We used this reference for development of data collection tool, since it was additional to other literatures reviewed and no other guideline developed after it.

8. The advantages and disadvantages of this study design

The advantage was being follow up study which stronger than cross-sectional/case-control studies

The disadvantage was being retrospective using secondary data

9. "Important modifications were made" Which kind of modifications were made and why?

10. Modifications made were due to incompleteness data from secondary sources like socio-demographic factors; marital status, occupations, pre-hospital factors; management given and others after pretesting of data collection tools.

11. "398 road traffic accident victims" there is a discrepancy in the previous reported 403 included patients to this statement. Where are the 5 missing patients?

The final sample size was 403, but the study conducted in 398 patient cards. the rest 5 patient cards were excluded due to incomplete data. 

12. "The majority, 89.45% of the victims, had no comorbid conditions. (Table 1)" What kind of comorbid conditions where these? Are they contributing to the outcome? Such as blood-thinners are pre-existing severe illnesses?

Comorbid conditions were pre-existing chronic diseases like diabetes mellitus, hypertension, HIV/AIDS, cancers of different origins and others which had significant effect on the outcome variable 

13. "More than one-third (36.93%) of the victims have had poly-trauma; whereas 35.17% were injuries to multiple body regions". Please clearly state the difference here again, so the reader can follow.

The first one was about occurrence of poly-trauma (35.17%); yes there was error in typing

14. "About 64.82% of the road traffic accident victims had to bleed"

They were bleeding at the time of arrival to the hospital or prior to that after accident had been occurred. 

15. Please rephrase the sentence.

16. "The most commonly performed management procedure was fixation (30.90%)"

More than a quarter (30.90%) of victims had got fixation as a management procedure

17. To get a better understanding of the mortality rate, it is important to differentiate between the first surgical treatment of the patient from the external referring hospital and the receiving hospital. 

For example, when were the interventions carried out? Since the majority of the patients were transferred from different hospitals, the need to be a distinction of when the surgical interventions were carried out.

Really, this was very important concern for our study, but it was very difficult to get this information from secondary data for all victims. Therefore, it could be the limitation of our study.

As it was follow-up study, we were interested to follow the victims after admission to discharge in the selected health facilities. So that it was impossible to follow outside the selected health facilities. 

18. In the described mortality rate, what was the leading cause of death? It is described in great detail, that the GCS values and a high impact on mortality. However accompanying injuries and leading cause of death is not clear to me.

As to our understanding, it was not the objective of this study and we tried to meet the objectives well.

19. "The current study revealed that the incidence of mortality among road traffic accident victims in the age group above 50 years was high, which is not consistent with studies undertaken in Yemen, Ghana, the D.R. Congo, and Ethiopia where the rate of death was higher among the age group below 50 years"[12, 20, 21, 25]

20. "Another possible reason might be that most of the study victims who arrive at these hospitals spent their time in another institution without getting adequate treatment."

"Another possible reason might be most of the study victims who arrive at these hospitals spent their time in another institution without getting adequate treatment."

According to our study, nearly three fourth (72.61%) of road traffic accident victims have had referred from another health facility which might cause delay without getting adequate management. 

Comments to the author section (Reviewer #1)

1. Here median is sometimes presented with SEM (standard error of median), where IQR should be stated. Further some of SEM presented doesn’t make sense. Please reevaluate and correct where necessary.

A total of 398 patients were followed for a range of 24 to 720 hours, and median follow-up time was 120 +/- 144 hours. Here, the median was 120 hours and IQR was 144 hours.

If any discrepancy in the data set, we can see it! Really we appreciate the critical observation of reviewer. 

2. Could the authors please shortly explain the rationale for using patient hours of observation as the metric for conveying mortality rate?

Our analysis plan was using survival analysis; so it needs time to event data set and the event was death.

3. I would recommend adding this data in table form to make it more easily understood.

We omit it from table to avoid duplicate presentation

4. In the results section, the data from Table 6 should be presented.

We think table 6 was not important here, really we thanks for detail observation

5. Further, in Table 1, please remove “having removal form” as this is redundant since referral status was also listed.

It was ‘having referral form’; the first part or referral status was added if some of them might not have referral form.

---

## [Editor Report · Decision Letter 1]

21 Dec 2023

Incidence and predictors of mortality among road traffic accident victims admitted to hospitals at Hawassa city, Ethiopia

PONE-D-23-20417R1

Dear Dr. Afacho,

We’re pleased to inform you that your manuscript has been judged scientifically suitable for publication and will be formally accepted for publication once it meets all outstanding technical requirements.

The authors have addressed all recommendations of the reviewers.

Kind regards,

Hans-Peter Simmen, M.D., Professor of Surgery

Academic Editor

PLOS ONE
---

## [Editor Report · Acceptance letter]

21 Feb 2024

PONE-D-23-20417R1 

PLOS ONE

Dear Dr. Afacho, 

I'm pleased to inform you that your manuscript has been deemed suitable for publication in PLOS ONE. Congratulations! Your manuscript is now being handed over to our production team.

Kind regards, 

on behalf of

Dr. Hans-Peter Simmen 

Academic Editor

PLOS ONE